# Drug-tunable multidimensional synthetic gene control using inducible degron-tagged dCas9 effectors

Dirk A. Kleinjan[1], Caroline Wardrope[1], Si Nga Sou[1] & Susan J. Rosser[1,2]

The nuclease-deactivated variant of CRISPR-Cas9 proteins (dCas9) fused to heterologous transactivation domains can act as a potent guide RNA sequence-directed inducer or repressor of gene expression in mammalian cells. In such a system the long-term presence of a stable dCas9 effector can be a draw-back precluding the ability to switch rapidly between repressed and activated target gene expression states, imposing a static environment on the synthetic regulatory circuits in the cell. To address this issue we have generated a toolkit of conditionally degradable or stabilisable orthologous dCas9 or Cpf1 effector proteins, thus opening options for multidimensional control of functional activities through combinations of orthogonal, drug-tunable artificial transcription factors.

[1] UK Centre for Mammalian Synthetic Biology at the Institute of Quantitative Biology, Biochemistry, and Biotechnology, SynthSys, School of Biological Sciences, University of Edinburgh, Edinburgh, EH9 3BF, UK. [2] Institute for Bioengineering, University of Edinburgh, Faraday Building, The King's Buildings, Edinburgh, 2 EH9 3DW, UK. Correspondence and requests for materials should be addressed to S.J.R. (email: Susan.Rosser@ed.ac.uk)

R ecent advances in our ability to engineer artificial transcription factors have brought the capability to design novel synthetic gene expression programs within reach. In particular the clustered, regularly interspaced, short palindromic repeats (CRISPR)-associated protein Cas9 has proved a powerful scaffold for the creation of a suite of transcription factors with desired functionalities[1, 2]. Cas9 is an RNA-guided DNA binding protein that can be directed efficiently to any genomic sequence through the use of short single-guide RNAs (sgRNAs), where it makes double-strand DNA breaks via its two endonuclease domains[3]. Inactivating missense mutations in both domains have created a nuclease-dead Cas9 protein (dCas9) that can be given new activities by fusion to functional protein domains, commonly at its C-terminus. For instance, fusion with the transcriptional activation domain VP64 turns dCas9 into a potent transcriptional activator, while attachment of the KRAB or SID4x domains produces a transcriptional repressor[4–6]. Without further modification the dCas9 protein can interfere with the process of transcription, when targeted to the transcriptional start site of genes in a process called CRISPRi[4, 7]. Unfused dCas9 is also used

in the second generation dCas9 transcription system, the SAM system[8], where instead the single-guide RNA (sgRNA) bears an RNA aptamer that recruits its cognate aptamer-binding protein fused to the desired functional domain.

A big advantage of the Crispr-Cas9 system is that a single dCas9 entity can be used to carry out its function at many different sites in the genome simultaneously, by supplying it with multiple small guide RNAs (sgRNAs). In contrast, in alternative systems such as TALEs and ZFNs the DNA binding specificity is encoded in the effector protein itself and targeting of multiple genomic sites would therefore require the introduction of a specific factor for each of them. However, as the dCas9 protein will readily combine with any supplied guide RNA in a cell, a clear drawback is that the combined use of different functionalities is troublesome and switching between different regulatory states, a highly desired requisite for the proper functioning of regulatory circuits, is precluded by the persistence of activating or repressing dCas9.

To address the need for a system that would allow user-controlled fine-tuning of dCas9 activity with regard to level and

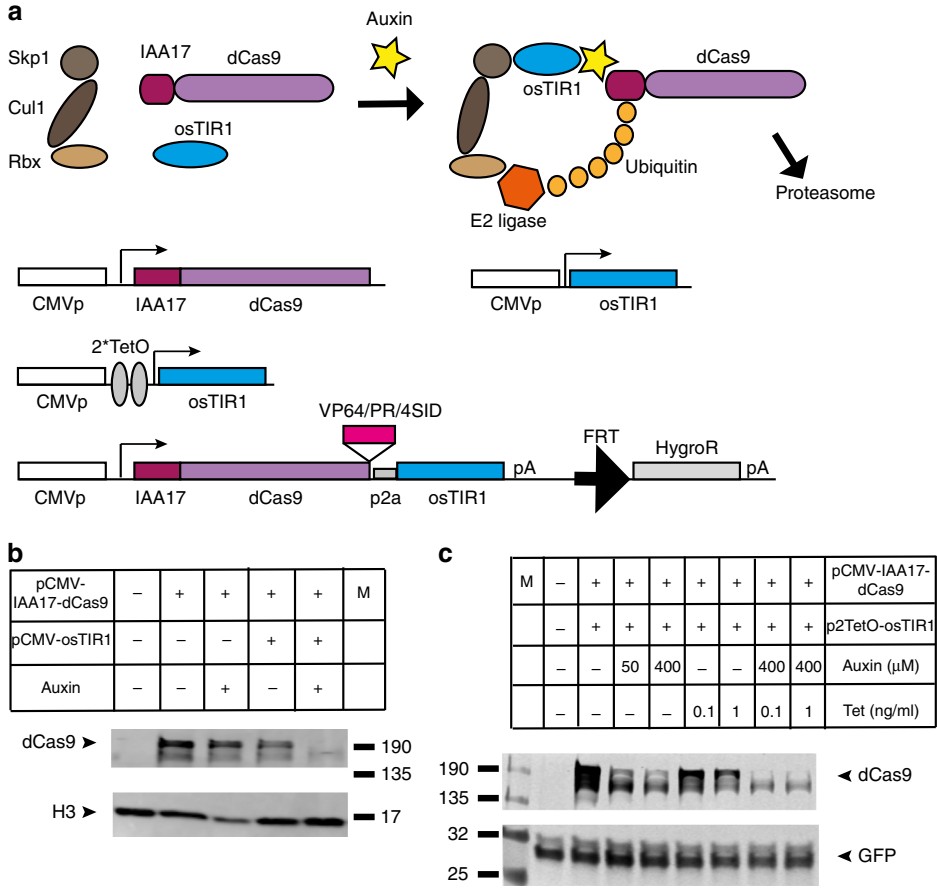

**Fig. 1** Small molecule inducible degradation control of dCas9 using the Auxin inducible degron. **a** Schematic overview of the AID-dCas9 system. The IAA17 peptide, which confers sensitivity to degradation upon addition of the plant hormone auxin, is fused to the N-terminus of dCas9. The cullin/skp/rbx components are present in mammalian cells, but auxiliary protein TIR1 has to be externally supplied. Upon addition of auxin a complex is formed with E2 ligase, resulting in poly-ubiquination and proteasomal degradation. IAA17-dCas9 and osTIR1 can be expressed from separate plasmids, allowing independent tetracyclin-inducible control of osTIR1 expression, or from a single combined construct suitable for Flp-mediated integration into FRT containing genomic insertion platforms. Effector domains of interest (e.g., VP64) can be fused to the C-terminus of AID-dCas9. **b** The IAA17 degron tagged dCas9 fusion protein is sensitive to degradation upon addition of auxin only in the presence of osTIR1. Western blot of HEK293FT cells transfected with the constructs shown in the table in the presence or absence of added auxin (400 μM) and probed with Cas9 antibody. Histone H3 antibody was used as loading control **c** Placing osTIR1 under control of a Tetracyclin (Tet)-inducible promoter results in degradation of IAA17-dCas9 only in the presence of both auxin and tetracyclin, effectively creating an AND or NAND gate depending on the C-terminal effector domain attached to dCas9. HEK293FT cells were transfected with pCMV-IAA17-dCas9 and pCMV-2TetO-osTIR1 and tetracycline or auxin were added as indicated in the table. A control GFP expression plasmid was co-transfected. Samples were probed on Western blot with Cas9 antibody. GFP antibody was used as loading control

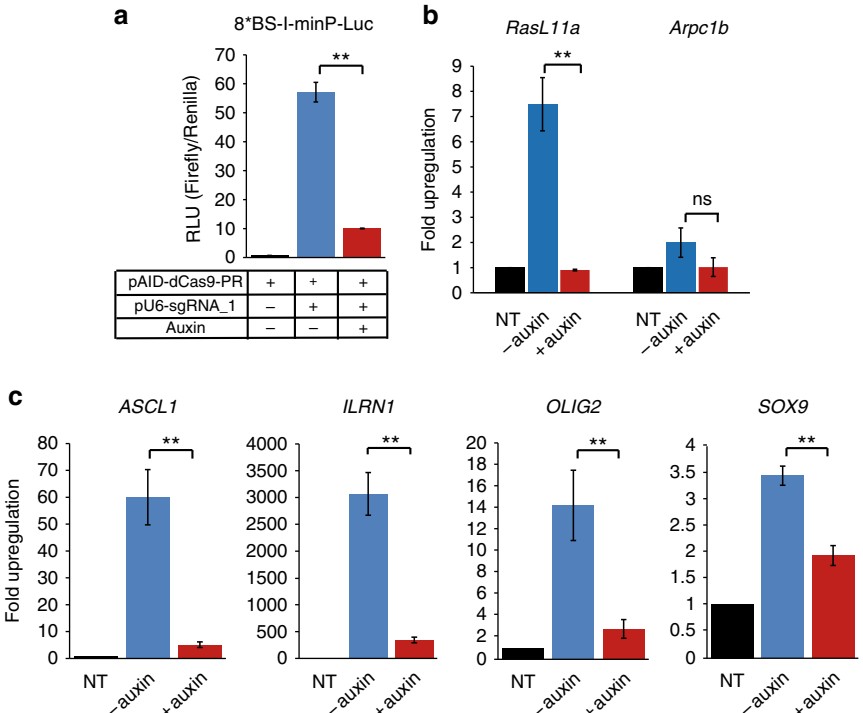

**Fig. 2** Auxin controllable functional activity of the AID-dCas9-PR synthetic transcription factor **a** Dual-luciferase assay on HEK293s transiently transfected with AID-dCas9-PR, an sgRNA expression construct (gRNA_1), a construct containing firefly luciferase under control of a minimal promoter with 8 binding sites that are recognised by gRNA_1 (p8*BS-I-minP-Luc) and a Renilla luciferase control plasmid. Auxin was added to the cultures where indicated. **b** A stably transfected CHO cell line with AID-dCas9-PR was transfected with guide RNAs targeting the promoter regions of RasL11a and Arpc1b. RNA levels after 48 h were assessed by rtqPCR. Upregulation was largely abrogated in the presence of auxin. **c** Upregulation of endogenous genes in HEK293 cells by auxin-inducible degron-tagged dCas9-PR is similarly affected by the presence of auxin. Expression levels of four genes, ASCL1, IL1RN, OLIG2 AND SOX9, representing a wide range of potential upregulation were assayed by rt-qPCR. Error bars denote ± s.d. (n = 3), (2-tailed t-test, (**) $P < 0.01$, (*) $P < 0.05$, (ns) not significant). Black columns, no sgRNA transfected (NT); Blue columns, no auxin added; Red columns, auxin added

timing (in addition to its target specificity bestowed by gRNAs) we aimed to generate a toolkit of drug-controllable degron tagged dCas9 variants. A degron is a peptide domain whose presence confers a greatly increased rate of protein degradation, and a variety of degrons have been described[9]. We employed two oppositely functioning small-molecule inducible degron systems, the Auxin-Inducible Degron (AID) and the ecDHFR destabilising domain (ecDDD), and tested their ability to control the stability of *Streptococcus pyogenes* dCas9. The AID degron system requires tagging the protein of interest with a domain (derived from the *Arabidopsis thaliana* IAA17 protein) that is sensitive to the plant hormone auxin[10]. In the absence of auxin, stability of the IAA17 degron tagged fusion protein is unaffected, but addition of auxin to the medium promotes interaction with an E3 ubiquitin ligase SCF complex containing the auxin perceptive F-box protein TIR1, which recruits an E2 ubiquitin conjugating enzyme leading to rapid poly ubiquination and degradation by the proteasome (Fig. 1a). While other parts of the SCF complex are endogenously present, application of the AID system in mammalian cells requires co-expression of the auxiliary protein TIR1, in particular a thermo-stable version from rice plants (osTIR1)[10, 11].

The ecDHFR destabilising domain (ecDDD) is an unfolded, structurally unstable domain derived from the *Escherichia coli* dihydrofolate reductase (DHFR) gene, which was evolved for enhanced instability by introduction of two additional missense mutations (R12Y/Y110I), which when fused to proteins of interest similarly targets the fusion protein for rapid proteasomal degradation[12]. In contrast to the AID the ecDDD does not require any additional factors to function in mammalian cells. The small-molecule drug trimethoprim (TMP) can bind and

stabilise the ecDDD in the fusion protein and prevent its proteosomal degradation in a concentration dependent manner[12].

In this study we generate a set of dCas9 family proteins fused with the AID and ecDDD degrons and characterise their capability for drug controlled protein destabilisation and functional tunability. We show that attachment of the AID does not impair the ability of dCas9-effector fusion proteins to regulate gene expression using both *S. pyogenes* and *S. aureus* dCas9, while addition of auxin induces their rapid and concentration dependent degradation. The ecDDD is less effective for destabilisation of dCas9 variants, but can be used to stabilise effector domain fusions of an aptamer binding protein (the MS2 coat protein) upon TMP addition, thus enabling multi-dimensional drug-tunable gene control via the use of AID-dCas9 in combination with aptamer containing guide RNAs.

## Results

**Auxin inducible control of dCas9.** We started by cloning the auxin dependent degron IAA17 at the N-terminus of a catalytically inactive *S. pyogenes* Cas9 (SpdCas9 (D10A, H480A)) in a mammalian expression construct. To allow for easier subsequent stable insertion into the genome of selected cell types via recombinase mediated genomic integration we generated a construct incorporating an FRT-site followed by a promoter-less hygromycinR gene cassette (Fig. 1a). To test whether the fusion protein is properly expressed and has acquired sensitivity to the addition of auxin to the medium we transfected HEK293FT cells and tested for the presence of IAA17dCas9 by western blot (Fig. 1b). As the AID system requires the auxiliary factor TIR1 we performed transfections with and without co-transfection of an

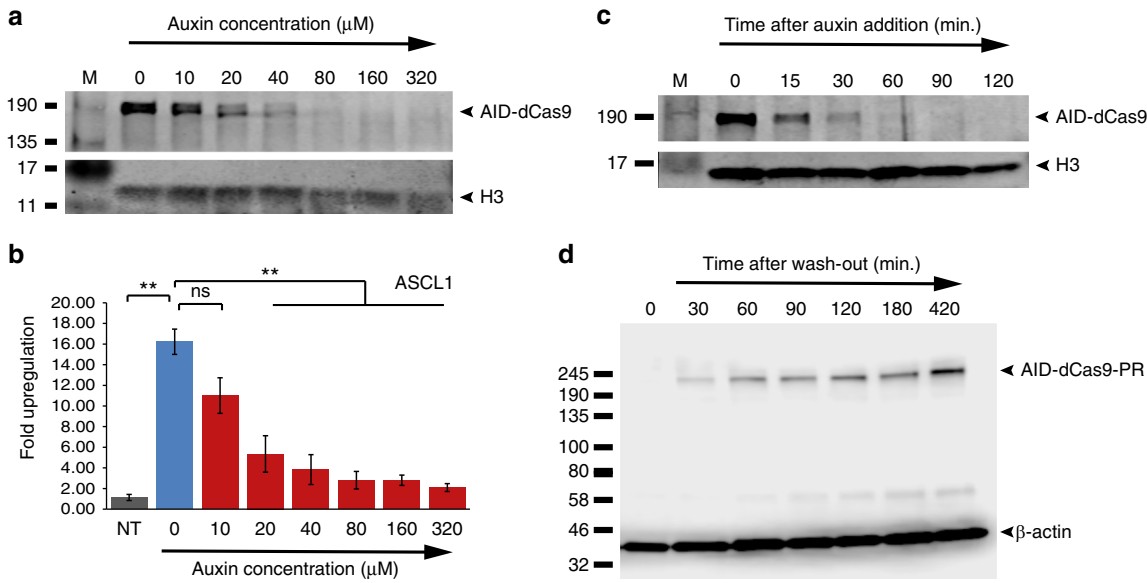

**Fig. 3** Concentration dependence and kinetics of the AID-dCas9-effector system. **a** Western blot on protein extracts from AID-dCas9 expressing HEK293 cells probed with Cas9 and histone H3 (control) antibodies showing the degradation efficiency of AID-dCas9 is dependent on the auxin concentration in the media. **b** Functional tunability of AID-dCas9 effectors. A stable AID-dCas9-PR cell line was transfected with a mix of guide RNAs targeting the *ASCL1* promoter region. Auxin was added to the concentrations indicated. mRNA induction levels decreased with increasing auxin concentration. Black columns, NT, no sgRNA transfected; Blue columns, no auxin added; Red columns, auxin added at indicated concentration. Error bars denote ± s.d. ($n = 3$), (two-tailed *t*-test, (**) $P < 0.01$, (*) $P < 0.05$, (ns) not significant). **c** Time course of the disappearance of AID-dCas9. Western blot of samples taken at the indicated time periods after addition of auxin. **d** Media wash-out and replacement with media without auxin results in detectable reappearance of AID-dCas9 within 30 min

osTIR1 expression construct. Expression of IAA17dCas9 reveals a stable protein of the expected size, which is unaffected by auxin addition in the absence of osTIR1. Co-transfection of osTIR1 in the absence of auxin produces a moderately destabilised IAA17dCas9 fusion protein, presumably due to the presence of a tiny amount of auxin in the culture media or a low degree of interaction between the IAA17 tag and osTIR1 even in the absence of auxin. Combined addition of auxin and osTIR1 co-transfection leads to efficient degradation of IAA17dCas9 (Fig. 1b).

**AND-gate logic with IAA17dCas9 and osTIR1**. The dependence for effective IAA17dCas9 removal on two separate factors (auxin and osTIR1 presence) suggested the system could be utilised as an AND-type logic gate. To test this option we placed the osTIR1 complementary DNA (cDNA) under the control of a tetracyclin inducible promoter. Cells were co-transfected with IAA17dCas9 and CMV-2TetO-osTIR1 and auxin or tetracyclin were added individually or in combination (Fig. 1c). Only in the presence of both drugs is the IAA17dCas9 protein efficiently degraded, demonstrating the feasibility of using the system as an AND or NAND gate depending on the effector domain fused to IAA17-dCas9.

**AID-dCas9 artificial transcription factors**. While the use of two plasmids to create a (N)AND-gate option can be useful, it is generally undesirable to require co-transfection of multiple constructs; therefore we designed a plasmid construct for simultaneous co-expression of IAA17-dCas9 and osTIR1 through linkage via a p2a peptide bridge (cassette subsequently referred to as AID-dCas9). A unique XmaI restriction site at the C-terminus of the dCas9 cDNA moiety allows for the introduction of selected effector domains. We produced versions with VP64 and PR (p65-rTa) transactivation domains as well as the SID4x repression domain[6, 13] (Supplementary Fig. 1a, b). Next we tested the ability of the degron tagged dCas9-PR to induce gene expression of

target genes. We co-transfected the AID-dCas9-PR with a reporter construct expressing luciferase under control of a minimal promoter containing eight recognition sites for an artificial guide RNA (gRNA-1), and observed a strong luciferase signal in the presence of gRNA-1. No luciferase activity was detected in the absence of the gRNA-1. Addition of auxin resulted in a sharp drop in luciferase output (Fig. 2a). To test the capacity for induction of endogenous genes we co-transfected AID-dCas9-PR with a mix of guide RNAs targeting the *ASCL1* and *SOX9* genes. Both genes show clear upregulation of expression, which is strongly reduced in the presence of auxin (Supplementary Fig. 2a, b), demonstrating the potential of AID-dCas9-PR as a drug-controllable transcriptional activator.

We proceeded to produce stable cell lines with the AID-dCas9-PR construct via Flp mediated integration into transcriptionally competent genomic landing sites in HEK293Trex-FlpIn and CHO-K1 derived cell lines. Functional activity of the integrated AID-dCas9-PR transactivator was confirmed via transfection of fluorescent or luciferase reporters under control of artificial gRNA binding site containing promoters and co-transfection of the corresponding guide RNAs. Addition of auxin to the medium severely reduced reporter output, to below detectable level in the case of the fluorescent reporters (Supplementary Fig. 2c). Similarly, targeting a number of endogenous genes via transfection of mixes of expression plasmids for 3 or 4 sgRNAs to sites within the same promoter region (*RasL11a* and *Arpc1b* in CHO cells (Fig. 2b) and *ASCL1*, *IL1RN*, *OLIG2* and *SOX9* in HEK293 cells (Fig. 2c)) resulted in clear transcriptional upregulation, which was markedly reduced in the presence of auxin.

**Tunability and kinetics of AID-dCas9**. Gene expression programmes of natural genes often involve a tightly regulated spatio-temporal on- and off-turning, in some cases requiring fast switching between expression states, in addition to strict dosage control. We tested the dosage dependence of AID-dCas9-PR by

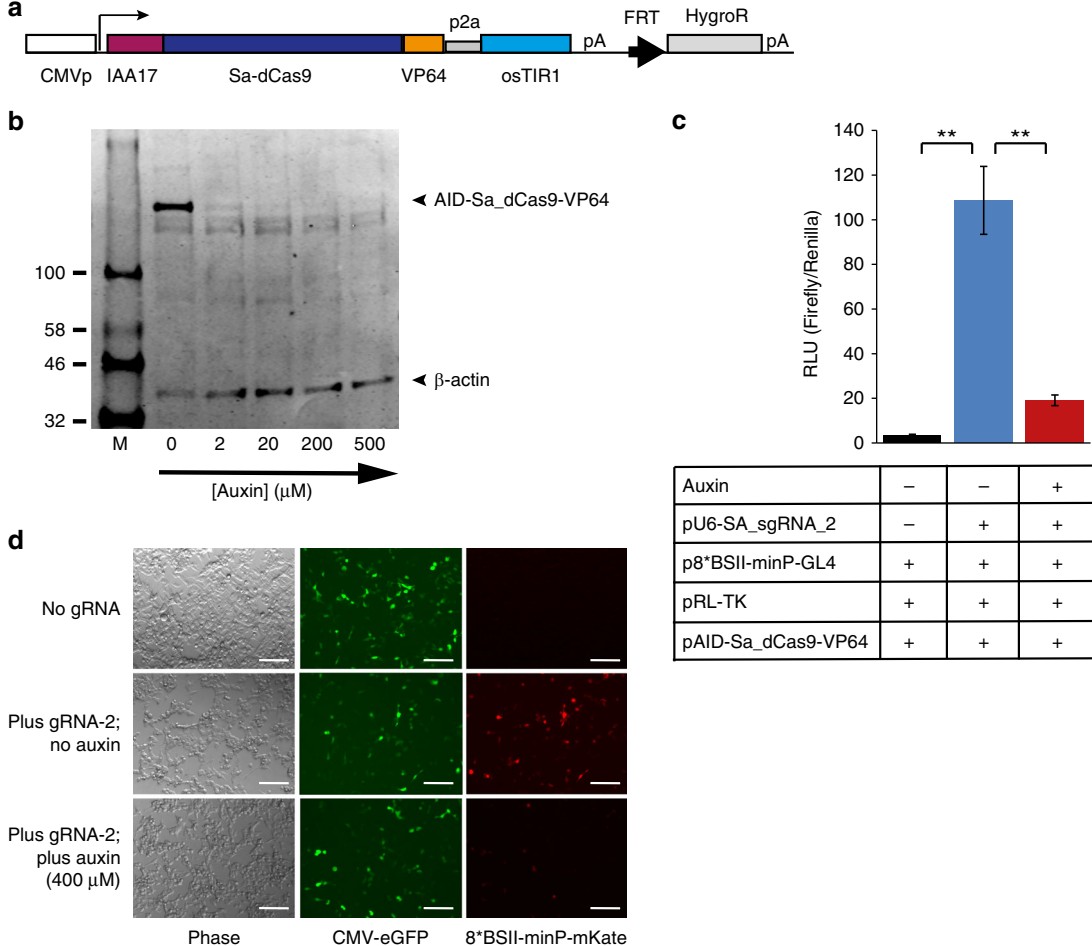

**Fig. 4** The AID system functions with stably integrated *Staphylococcus aureus* dCas9 (Sa-dCas9) fused to the VP64 transactivation domain. **a** Schematic representation of the AID-Sa-dCas9-VP64 construct. **b** Western blot of HEK293-Trex-FlpIn cells stably expressing AID-Sa-dCas9-VP64 and treated with increasing concentration of auxin as indicated. **c** Dual luciferase assay on stably expressing AID-Sa-dCas9-VP64 cells transfected with a luciferase reporter driven from a minimal promoter containing eight binding sites for an *S. aureus* compatible guideRNA (p8BS-II-minP-GL4), a Renilla luciferase control plasmid containing the ubiquitous Thymidine kinase promoter (pRL-TK) and an *S. aureus* sgRNA (pU6-Sa_gRNA-2) expression construct. Black column, no sgRNA transfected; Blue column, no auxin added; Red columns, auxin added (400 μM). Error bars ± s.d. ($n = 3$). (**\*\***) $P < 0.01$ two-tailed *t*-test. **d** Fluorescent reporter expression is abrogated in the presence of auxin in cells stably expressing AID-Sa-dCas9-VP64. Scale bar, 100 μm. All samples were transfected with a constitutive GFP expression construct, an mKate construct with a minimal promoter containing eight binding sites for an *S. aureus* guideRNA target sequence (p8BS-II_minP-mKate), and an *S. aureus* sgRNA (pU6-Sa_gRNA-2) expression construct (middle and bottom rows only), in the absence (top and middle rows) or presence of auxin (400 μM; bottom row)

assaying protein stability under increasing auxin concentration, showing an inverse correlation between auxin concentration in the medium and residual presence of the protein (Fig. 3a). This allows tunability of AID-dCas9-effector function as demonstrated by auxin concentration dependent upregulation of the endogenous *ASCL1* (Fig. 3b), *IL1RN* and *OLIG2* genes (Supplementary Fig. 3a, b). messenger RNA (mRNA) levels of dCas9-PR itself are unaffected by auxin (Supplementary Fig. 3c). Next we assayed the kinetics of the AID-dCas9 system by monitoring its disappearance over time after addition of Auxin (Fig. 3c), and its reappearance following wash-out of the drug (Fig. 3d). Cells were grown for 24 h to allow build-up of the protein. After addition of auxin samples were collected at defined intervals. Clearance of the pool of AID-dCas9-PR protein took 1–2 h, with an estimated half-life of <15 min, in accordance with reports on other AID tagged proteins[11, 14] (Fig. 3c). Next, cells were grown in the presence of auxin for 24 h before being replaced with medium lacking the drug. Detectable levels of the protein reappeared within 30 min, increasing back to maximum level over the next 3–7 h (Fig. 3d).

**Orthogonal, auxin degradable dCas9 variants.** Natural gene networks often rely on fast switching interactions of multiple transcriptional activators, repressors and other chromatin modifying factors with cohorts of responsive target promoters. To enable recreation of such behaviours in artificial systems, allowing rapid and diverse, independently controllable functional activity, we sought to expand our system by developing a set of orthogonal, auxin-degradable synthetic transcription factors. We first replaced the *S. pyogenes* dCas9 cDNA in our AID vector with a small multiple cloning site upstream of an HA-tag, into which we inserted several alternative, orthogonal CRISPR effector proteins (*S. thermophiles* 1 dCas9, *S. aureus* dCas9, A.s Cpf1, L.b Cpf1 and F.n Cpf1)[15, 16]. Western blots of HEK293 and CHO-K1 cells transfected with this set revealed the auxin induced degradation potential of these orthogonal AID-tagged factors (Supplementary Fig. 4a, b). Remarkably the AID-Cpf1 factors showed a clear difference in response to the drug between the cell lines. Although the reason is presently unclear, this could be due to differences in the expression levels of some auxiliary factors (e.g., SCF complex subunits) in combination with reduced accessibility of the IAA17

tag at the N-termini of the Cpf1 proteins. Disappearance of *S. aureus* dCas9 in both cell lines upon treatment suggested SadCas9 as a promising orthogonal auxin-controllable transcription factor (Supplementary Fig. 4a, b). We replaced the HA-tag with a VP64 transactivation domain in the AID-Sa-dCas9 construct and generated a stably expressing line in HEK293 cells (Fig. 4a). Auxin dependent protein stability of AID-SadCas9-VP64 in this line was

confirmed by Western blot (Fig. 4b). Using an *S. aureus* sgRNA specific for a repeated site in the promoter of a luciferase reporter construct we demonstrate functional activity of AID-SadCas9-VP64, which largely disappears in the presence of auxin (Fig. 4c). Similarly the induction of an mKate reporter under control of a multiple *S. aureus* sgRNA target site-containing minimal promoter by AID-SadCas9-VP64 in the presence of the sgRNA is

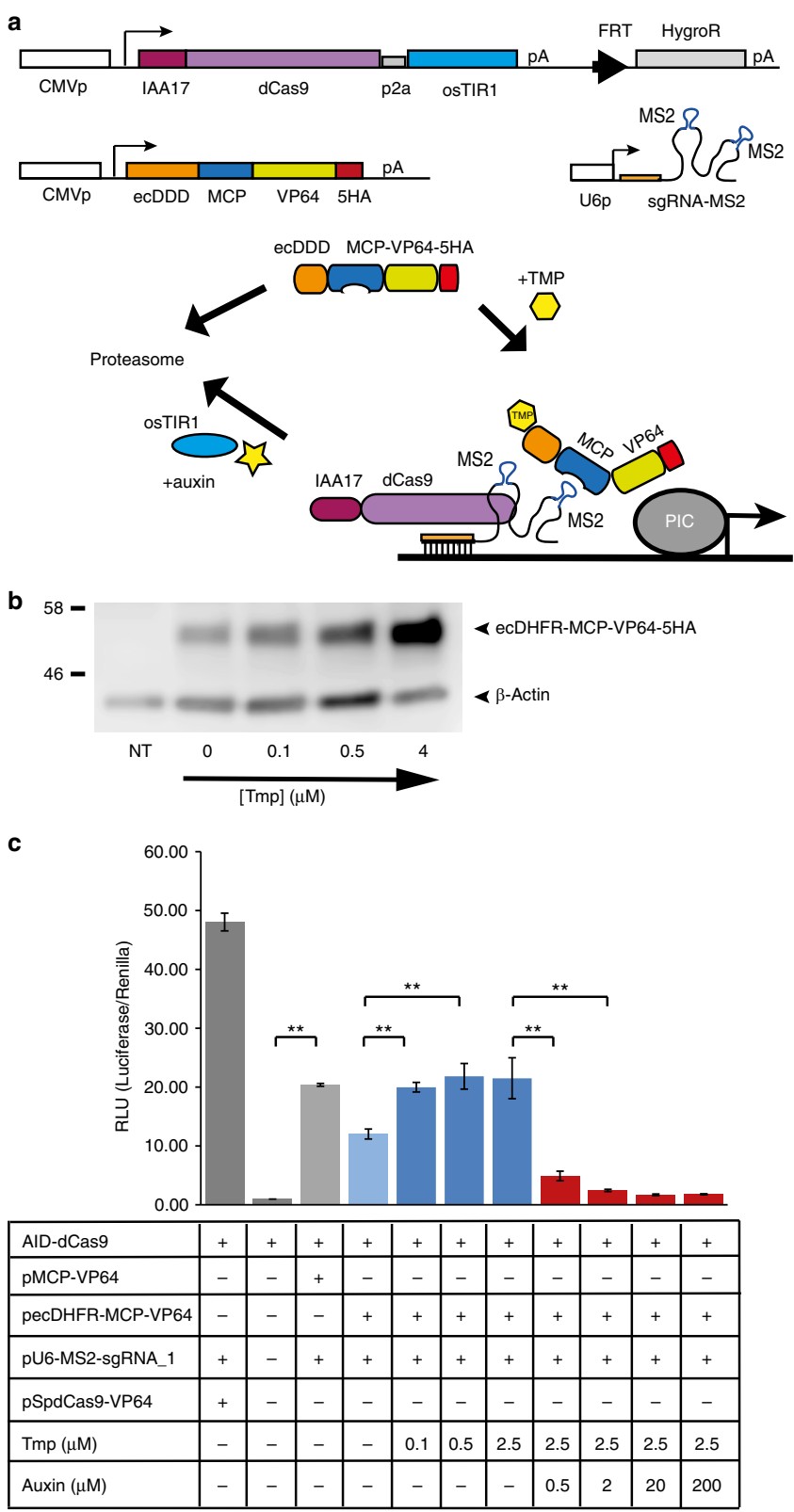

| AID-dCas9 | + | + | + | + | + | + | + | + | + | + | + |
|---|---|---|---|---|---|---|---|---|---|---|---|
| pMCP-VP64 | − | − | + | − | − | − | − | − | − | − | − |
| pecDHFR-MCP-VP64 | − | − | − | + | + | + | + | + | + | + | + |
| pU6-MS2-sgRNA_1 | + | − | + | + | + | + | + | + | + | + | + |
| pSpdCas9-VP64 | + | − | − | − | − | − | − | − | − | − | − |
| Tmp (µM) | − | − | − | − | 0.1 | 0.5 | 2.5 | 2.5 | 2.5 | 2.5 | 2.5 |
| Auxin (µM) | − | − | − | − | − | − | − | 0.5 | 2 | 20 | 200 |

abrogated upon addition of auxin (Fig. 4d). This indicates the potential of AID-SadCas9-VP64 as a potent orthogonal, drug-controllable artificial transcription factor.

**dCas9 proteins with the ecDHFR destabilising domain**. While orthogonal AID dCas9 proteins can be tagged with different effector domains to carry out a diverse range of activities, increased versatility would be gained by having these functions under control of independent degron domains regulated by different small-molecule drugs. The ecDHFR DD (ecDDD) is an inducible degron that functions in opposite direction to the AID degron, i.e., its intrinsically unstable fusion protein is stabilised by addition of the drug trimethoprim (TMP) to the medium[12] (Supplementary Fig. 5a). We first attached the ecDDD to *S. pyogenes* dCas9. Though some residual protein was detectable by Western blot, stability of the fusion protein was greatly improved by addition of TMP (Supplementary Fig. 5b). Next we constructed ecDDD tagged versions of the set of orthogonal dCas9 or Cpf1 proteins (*S. thermophiles* 1 dCas9, *S. aureus* dCas9, As.Cpf1, Lb.Cpf1 and Fn.Cpf1). With the exception of a moderate effect on SadCas9, the addition of the ecDDD degron did not appear to affect the stability of the set of orthogonal Cas9 or Cpf1 proteins when transfected in HEK293FT cells (Supplementary Fig. 5c). To investigate whether cell-type specific characteristics would affect degradation efficiency we next tested the effect of the degron-tag in CHO-K1 cells. CHO cells are an important resource in the biotechnology industry and a valuable target for systems that enable the drug controlled manipulation of gene expression output[17]. Interestingly the sensitivity of our panel of degron-tagged Cas9 or Cpf1 proteins to proteosomal degradation is markedly enhanced in CHO cells compared with 293 cells, showing a clear difference in drug dependent stability of the fusion proteins for most of the constructs in both the AID and ecDDD degron tagged sets (Supplementary Fig. 5d).

**Stereo-tuner dCas9**. While the limited instability-conferring performance of the ecDDD on the set of orthogonal dCas9/Cpf1 proteins might allow some useful modulation of functional output we reasoned that the relatively large size of the dCas9-effector proteins may affect its sensitivity to the ecDHFR degron. We therefore shifted our focus to applying the ecDDD degron to the smaller MS2 core protein (MCP) used as effector-supplying factor in the CRISPR SAM system[8], with the aim to achieve enhanced tunability of dCas9-effector activity through double degron tagging (Fig. 5a). In the SAM system the guide RNA scaffold is engineered to contain two MS2 aptamers, which are recognised by the MCP protein to which various effector domains (e.g., VP64 or p65AD-Hsf1) can be attached. We inserted an MCP-VP64 module into our ecDHFR expression vector to generate a TMP stabilisable ecDHFR-MCP-VP64 fusion protein. While we find a low level of ecDHFR-MCP-VP64 present even in the absence of

TMP, addition of the drug effects stabilisation in a concentration-dependent manner (Fig. 5b). We show that attachment of the ecDHFR degron does not affect the ability of the fusion protein to induce transcriptional activation in conjunction with AID-dCas9 (Fig. 5c). Using our luciferase assay we show that addition of TMP and/or auxin to the culture medium can up and down regulate the expression of luciferase from a responsive promoter (Fig. 5c). Tagging of a different aptamer binding protein, utilising the PP7 aptamer, with the ecDDD degron was recently also shown to be effective for tunable transcriptional activation[18]. Here we combine stabilisation of the ecDDD tagged MCP-effector module with the tunable degradation of the AID-dCas9 protein. We call this system 'stereo-tuner dCas9', and envisage that combination of the domain-less AID-dCas9 DNA binding module with various ecDHFR-MCP-effector domain plasmids will allow precisely targetable functional activities that can be finely tuned in opposite directions by two independent small molecule drugs.

**Discussion**

A major aim in synthetic biology is the development of versatile and programmable genetic regulators that enable strict control of gene expression programmes. The ability to regulate the stability of such synthetic transcription factors through application of small molecule drugs will enable a more precise control over the timing and dosage of their activity and enhance the scope of their application in basic science and therapeutic settings. We demonstrate here that the auxin inducible degron and to lesser extent the ecDHFR degron are effective drug-inducible regulatable domains for *S. pyogenes* dCas9. The AID degron also functions with selected orthogonal dCas9 and Cpf1 proteins, in particular with *S. aureus* dCas9, while the ecDHFR degron allows a more subtle modulation of their expression levels, or of the level of an aptamer binding protein linked with an effector domain. We envisage that tailored combinations of the orthogonal, degron tagged effector variants from this toolkit will enable the precise targeting of a variety of functional domains to their desired destination in a precise, timing and dosage controlled manner. Furthermore, the range of inducible dCas9/dCpf1 factors can be expanded with existing tools such as a recently reported tamoxifen-inducible dCas9[19]. In that system however, the dCas9-ER is not degraded but sequestered in the cytoplasm until application of the drug induces passage into the nucleus, and thus it does not have the benefit of the increased, controllable turn-over rate of the AID-dCas9 system that can enable faster switching between different functional activities. Moreover, the use of tamoxifen/estradiol as inducer may be less suitable in certain situations due to the potential for unwanted side effects of the drug, which is unlikely to be an issue with the plant hormone auxin. The availability of a range of small molecule-inducible factors controlled via different modes of action will enable the

**Fig. 5** Stereo-tuner dCas9 system. **a** Schematic overview of the system. An effector-less dCas9 is tagged with the auxin-inducible degron allowing inducible degradation upon addition of auxin. Two MS2 aptamer sequences are included in the single-guide RNA scaffold, as used in the Crispr/SAM system[8]. A transcriptional effector domain (e.g., VP64) is brought in by the MS2 aptamer-interacting MS2 core protein (MCP) which is fused to the ecDHFR destabilisation domain (ecDDD). Addition of the small molecule drug trimethoprim (TMP) improves the stability of naturally destabilised ecDHFR fusion proteins. PIC, Polymerase II initiation complex. **b** Western blot with HA antibody showing TMP concentration dependent stabilisation of the ecDHFR-MCP-VP64-5HA protein. β-actin antibody is included as loading control. **c** Dual-luciferase assay using a firefly luciferase reporter driven from a minimal promoter with eight copies of a guide RNA target sequence (gRNA-1) in an effector-less AID-dCas9 expressing cell line. A ubiquitous Renilla luciferase construct was co-transfected as internal control. In the presence of the MS2 aptamer bearing gRNA-1, luciferase activity is produced by transfection of a dCas9-VP64 plasmid (first lane) or by the co-expression of the endogenous AID-dCas9 with the pecDHFR-MCP-VP64 effector module, and can be enhanced by TMP administration. Addition of auxin to the medium, affecting IAA17-dCas9 stability, abrogates luciferase activation. Grey columns, controls; Light blue column, no TMP, no auxin; Dark blue columns, TMP added; Red columns, auxin added. Error bars denote ± s.d. (*n* = 3) (two-tailed *t*-test, (\**) *P* < 0.01)

design of tailored combinations of dCas9 effectors for specific applications. Another potential advantage of the AID-dCas9 system is its dependence on the presence of both auxin and osTIR1 expression that can be exploited by placement of the osTIR1 gene under the control of spatio-temporal or condition-dependent promoters, thus enabling tissue-specific abrogation of dCas9-effector activity. Even though in this study the auxin inducible degron was attached to nuclease dead Cas9 (dCas9) the same drug-inducible degradability could be extended to the nuclease active version of Cas9[20]. In situations were potential off-target cutting could be an issue[21] it may be beneficial to be able to limit the period during which active Cas9 is present within the cell.

In conclusion, we believe that the AID-dCas9 system and its extension with the ecDDD-MCP aptamer binding module described here with its small molecule-inducible tunability adds a powerful functionality to the rapidly expanding range of dCas9 effectors in the synthetic biology toolkit.

## Methods

**Cell culture**. HEK293FT cells (Invitrogen R70007) were grown at 37 °C in a 5% $CO_2$ atmosphere in Dulbecco's modified Eagle's medium (DMEM)(Gibco; Life Technologies) supplemented with 10% FBS (Gibco; Life Technologies), 4 mM glutamine and 1% penicillin-streptomycin (Gibco; Life Technologies). CHO-K1 cells were grown in Ham's F12 medium with 10% FBS (Gibco; Life Technologies), 4 mM glutamine and 1% penicillin-streptomycin (Gibco; Life Technologies) at 37 °C and 5% CO2. For transient transfections cells were seeded at $0.5 \times 10^5$ cells per well in 24 well plates and transfected the following day with an appropriate amount of each plasmid in Opti-MEM reduced serum medium (Life Technologies) using Lipofectamine 3000 (Invitrogen). sgRNAs were produced in the cell from a U6 promoter construct into which 20 nucleotide guide sequences were cloned upstream of the appropriate species-specific sgRNA scaffold. GuideRNA recognition sequences used in upregulation experiments were chosen from the literature or designed using the Benchling Crispr design tool (https://benchling.com/crispr) and are listed in Supplementary Table 1 (Supplementary information). For upregulation of endogenous genes the constructs for 3 or 4 sgRNAs against different sites within the target gene promoter were combined to make a guideRNA mix for that gene.

For the experiment involving tetracyclin induction of pCMV-2TetO-osTIR1, HEK293Trex-FlpIn cells (Invitrogen) were grown in DMEM/Glut/PenStrep with 10% Tet system approved FBS (Clontech 631107).

Stable cell lines expressing the AID-dCas9-effector-osTIR1 cassette were generated using the 293Trex-FlpIn cell line (Invitrogen) or an in-house CHO cell line (CHO-B4; CHO-K1 derived) with a randomly integrated SV40 promoter-FRT site-Neomycin resistance gene containing cassette in its genome. Cells were transfected with the relevant construct containing an FRT site immediately upstream of a promoter-less Hygromycin resistance gene and a Flp recombinase expression plasmid (pCaggs-Flpe). Hygromycin was added 24 h post transfection at 125 μg ml$^{-1}$ for 293Trex-FlpIn cells or 350 μg ml$^{-1}$ for CHO-B4 cells.

CHO-K1 cells were obtained commercially (ATCC catalogue number CCL-61). Blasticidin and zeocin were periodically added to the growth medium of 293Trex-FlpIn cells to maintain selective pressure. Cells were monitored for mycoplasma contamination using the LookOut mycoplasma PCR detection kit (Sigma MP0035).

For functional assays drugs (0.5 mM auxin (Sigma I5148) in ethanol; 5 mM TMP (Cayman Chemical 16473) in DMSO) or vehicle were added 1 h prior to transfection and cells were harvested after 48 h. For tunability assays auxin was used at the lower concentrations indicated in the figures. For western blots, unless where indicated otherwise, drugs or vehicle were added 24 h following transfection and cells were collected at 48 h after transfection. Recoverability from auxin induced degradation was tested by growing cells in 0.5 mM auxin for 24 h, at which point the media was replaced with fresh media without auxin. Cells were grown in this fresh media for the indicated time period before being harvested.

**Plasmids**. The plasmids used in this study were constructed via direct subcloning or PCR amplification of genes and functional domains from a variety of donor plasmids obtained from the Addgene plasmid repository, including Addgene #47106 (gift from Charles Gersbach), #55195 (gift from Timothy Lu), #63798 (gift from George Church), #64113 (gift from Thoru Pedersen), #62322 (gift from Wendell Lim and Stanley Qi) and #61594, #61424, #69982, #69976, #69988 (gift from Feng Zhang). Further plasmids used were pcDNA5TO (Invitrogen), and pNHK36 and pMK43 (Riken BioResource Center, Japan). Other domains were synthesised as gBlocks (IDT).

**RNA extraction and quantitative RT-PCR**. Transfections were carried out in triplicate. Cells were collected by brief trypsinisation, centrifugation and resuspension in RNA lysis buffer. RNA extraction was performed with the EZNA Total RNA Kit 1 (Omega Biotek) according to the manufacturer's protocol. In total 2 mg of total RNA was DNAseI treated, followed by reverse transcription using SuperScript III (Invitrogen) using random hexamer primers. Triplicate qPCR reactions for each sample were run in a StepOnePlus real-time PCR machine using PowerSybrGreen qPCR-mix (Life Technologies). Primer sequences used are listed in Supplementary Table 1 (Supplementary information). Data were analysed by the ΔΔCt method, normalised to GAPDH or RCN1 expression and reported as the mean ± s.d. of three biological replicates averaged over three technical replicates. Statistical analysis was done as standard on the biological triplicates using a two-tailed Student's t-test

**Western blot analysis**. Cells grown in the presence or absence of auxin or TMP were lysed in RIPA buffer (50 mM Tris pH7.5, 150 mM sodium chloride, 0.1% sodium dodecyl sulphate, 0.5% sodium deoxycholate, 1% Triton-X100) with complete protease inhibitors (Roche). Lysates were centrifuged and transferred to a fresh tube. Normalised amounts of protein were loaded onto a 4–15% acrylamide gradient gel (Bio-Rad Mini Protean TGX) and run in Tris-glycine running buffer (25 mM Tris, 192 mM glycine, (20% (v/v) methanol (pH 8.3)). Gels were electro blotted onto PVDF membrane in CAPS transfer buffer (10 mM 3-(cyclohexylamino)-1-propanesulfonic acid, 10% methanol (pH 11.0)). In some cases gels were blotted overnight on nitrocellulose membrane in Tris-glycine blotting buffer (25 mM Tris, 192 mM glycine). Membranes were blocked overnight at 4 °C in 5% milk powder (Marvel) in PBS-T (PBS, 0.1% Tween-20), before being incubated for at least 4 h with primary antibody against S.pyogenes Cas9 (EpiGentek Cas9 polyclonal A-1111 or Clontech GuideIT Cas9 polyclonal 632607 (1:3000)), anti-HA (Sigma HA-7; 019K4833 (1:6000)) or S.Aureus Cas9 (Diagenode C15310259 (1:3000)). Anti histone H3 (Abcam Ab1791 (1:10,000)), anti-GFP (Clontech Living Colors monoclonal JL-8, 632380 (1:6000)) or HRP-conjugated anti-beta-Actin (Sigma monoclonal BA3R, MA5-15739-HRP (1:6000)) antibodies were used as loading controls. After washing in PBS-T membranes were incubated with HRP-conjugated secondary antibodies for > 1 h. Signal was detected using WesternSure premium chemiluminescence substrate (LiCor 926–95000) on a LiCor c-Digit instrument. For imaging on the LiCor CLX blots were incubated with LiCor IRDye secondary antibodies: IRDye 800CW donkey anti-mouse 926–32,212 (1:10,000), and IRDye 680LT donkey anti-rabbit 926–68023 (1:10,000). Images were captured in Image Studio Lite 3.1.4 (LiCor biosciences) and processed in Adobe Photoshop. Full images are shown in Supplementary Fig. 6.

**Luciferase assays**. Cells were transfected with a firefly luciferase reporter construct harbouring multiple guideRNA recognition sites upstream of a minimal promoter, alongside an appropriate guideRNA expressing construct, a degron-dCas9-effector construct and a Renilla luciferase internal control plasmid. After 48 h cells were lysed in Passive Lysis buffer and Firefly and Renilla luciferase induced luminescence levels were measured for each sample using the dual-luciferase kit (Promega E1910) on a Modulus II microplate reader (Turner Biosystems).

**Fluorescence imaging**. Fluorescence images were captured on a Leica DM IL LED microscope with a Leica DFC3000G camera using Leica Application Suite X software.

**Data availability**. The authors declare that the data supporting the findings of this study are available within the article and its Supplementary information. Additional data will be available online at the Edinburgh University Datashare repository (doi:10.7488/ds/2121).

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

## Acknowledgements

This work was supported by grants from the BBSRC (BB/M018040/1 and BB/M018229/1) to S.J.R.

## Author contributions

D.A.K. and S.J.R. conceived the study and analysed the results; D.A.K. designed the experiments; D.A.K., C.W. and S.N.S. performed experiments; D.A.K. wrote the manuscript.

## Additional information

**Competing interests:** The authors declare no competing financial interests.

