## [Peer Review File · Nature Communications]

Reviewers' comments:

Reviewer #1 (Remarks to the Author):

The authors created a new Cas9 system capable of switching when induced using degron-tags. Using dCas9 and VP64 the CRISPR/Cas9 system can activate transcription of a gene of interest. This ability was coupled to degradation by auxin and the ecDHFR destabilizing domain. This new tool was able to reduce the activation by gRNA and dCas9-VP64 on genes upon addition of auxin. The ecDDD system was not compatible with dCas9, so the authors attached the destabilizing domain to MCP with limited success. Overall, auxin was used to rapidly modify levels of dCas9 in HEK293 and CHO cell lines. The manuscript provides details the use of protein tags to regulate the stability of dCas9 Effectors. In summary, this work is significant because it details an approach to improve the control of gene expression by dCas9, a widely-used technology. The use of multiple gene targets, Cas9 variants, and two simultaneous degrons represents a substantial body of work to support the manuscript. However, the impact of the work is greatly diminished by the lack of clarity in the manuscript and figures. The specific issues detailed below limit the reviewer's enthusiasm for the manuscript and may lessen most reader's appreciation for the work overall. Major comments for the authors

- On lines 84 and 85, the authors suggest that the small reduction in protein stability observed with the co-transfection of TIR may in the absence of auxin may be due to a "low level of auxin in the culture media". Since auxin is a plant hormone, this probably isn't the case.
 - On line 118, the authors detail how the addition of auxin reduced the fluorescent reporter readout to below background level, as shown in supplemental figure 2B. This doesn't seem to correlate with supplemental figure 2B, and it doesn't make sense as to why the degradation of a transactivator would reduce the luciferase readout to below background, as background should be the system without the transactivator.
 - The authors need to comment on why there are such pronounced differences in the performance of the degron-tagged Cas9 orthologs in different cell lines (HEK293 and CHO). Since they are discussing the use of this system to control gene programs, this may be something worth discussing, since other cell lines could benefit from the technology presented in this manuscript.
 - In Figure 1b- A western blot for dCas9 is shown, but no load control is shown. The authors should repeat the blot with a load control (b-actin), similar to figure 3B to make the claim that Cas9 co-transfection with tir1 effect degradation (lines 85-86).
 - Figure 2a- The load control for lane 6 (160 μ M) indicates a poorly loaded well and isn't appropriate for publication. The concentration is not absolutely necessary to make the authors claim, so either remove the lane or repeat.
 - Lines 130-138 need to be re-written for clarity because the results for figures 2d and 2e appear to be discussed twice.
 - GFP load control needs to be better described. Is the control expressed from the same plasmid or co-transfected?
 - The figure data would be more clear if the cell types are clearly labeled and transient data is separated or at least distinguished from the stable cell lines.
 - Clarify when sgRNA mixes are used.
 - Add guide RNA description to the methods. Were these chosen at random, predicted by software, or candidates from a larger group?
 - Figure 2b and 2c are described as fast or rapid with no time in the presented data. The reference is possibly meant to describe slope or this reference is confused with 2d and 2e. The authors need to clarify the referencing to figures.
 - Figure 2d shows and change in protein without a load control. A load control is required to show changes in protein content.
 - In the methods, there is no discussion of how the gRNAs are designed or of the mix gRNA experiments.
 - Perform 2-sample t-test on fluorescence and qRT-PCR data to test for significance.
- Minor comments for the authors
- This manuscript reads as if it was a first draft, many revisions will likely be needed before the

next submission.

- Minimize subject-verb inversion for clearer writing.
- Keep acronyms and naming consistent between figures, legends, and main text. One example is in supplementary figure 4c, pRL-TK and p8*BSL1-minP-GL4 are used. However, these acronyms can't be found anywhere else in the article and are never explained.
- Proofread for comma placement and check for comma splicing.
- Break up large noun clusters - i.e. line 50-51: ...user-controlled temporal and dosage fine-tuning...
- Discuss the CRISPR SAM system (Lines 176 through 181) in the introduction.
- It is unclear what you mean by "transactivation domain-less AID-dCas9". If it is just AID-dCas9 without a transactivation domain just call it AID-dCas9.
- Make sure figures are provided in the same order that they are provided in the text. In this manuscript figures 1 a-c are discussed, followed by figure 2, then figures 1 d-f. Either split figure 1 into two separate figures or change the order of the writing so that figure 1 is completed before moving on to figure 2.
- It's unclear why GFP is included in figure 1c.
- Place subfigures in figure 1f on the same axes.
- Line 107 which figure are the authors referencing?
- Line 119 incorrect statement.
- "Multi-directional" is not a clear term to use in the title of the paper.
- Misspelling of "HEK293" on line 78.
- ecDDD is first defined on line 65 as "ecDHFR destabilizing domain", but then the term "ecDHFR DD" is used on line 156. It is recommended that the authors pick one term and stick with it.
- On lines 171 and 172, supplemental figure 5 is referenced for the AID and ecDHFR degron tagged sets, but the AID set is in supplemental figure 3.
- For lines 185-187, the only dCas9 observed in the table in Fig 3c is the one that has a VP64, yet they refer to a "transactivation domain-less AID-dCas9".
- On lines 238 and 239, they switched up the letter of the figures that they are referencing.
- Figure 3A needs to be clearer, because it's not apparent that the MCP is associating with the MS2 domains, or that the IAA17 would interact with the auxin and osTIR1.
- In the first line of the table in Figure 3 C they didn't include the AID tag in the name.
- In Supplemental Figure 4 C, columns 3 and 4 appear to be the same in regards to the table.
- Improve the consistency of figure style e.g. some have titles on top, others inside the plot.
- Figures should be referenced as Fig.1 or figure1 in accordance with the journal style. For example, in line 107 the authors write (figure 2b) and in line 117 the authors write (fig. 1d).
- All figures with RT-qPCR data- Y-axis should be titled " $\Delta\Delta$ ct fold change" or similar
- Figure 1a- Label on osTIR1 (the white text) is not easily visible
- Figure 1d re-title the figure (remove underscores in all titles)
- Figure 2a - Auxin concentration is given the unit μ M instead of μ M
- Figure 3a This figure would be more helpful to the reader if the schematic depicted only the protein interactions, but with greater detail.
- Line 40-44 - one long sentence. Should maybe break it up somewhere.
- Line 48 - should have a comma after "circuits"
- Lines 50-64 - these lines are really dense and have a lot of words that I needed to look up (i.e. degron, ecDHFR, TIR1).
- Line 63 - co-expression of the proteins in a mammalian system may lead to drawbacks.
- Lines 65-72 - a better explanation paragraph than the previous one. Easier to follow
- Line 76 - FRT-site containing pcDNA5 (?)
- Line 78- Change HEK392 to HEK293
- Line 107- Figure 2b is discussed before figure 1d,e,f.
- Line 110 - Supplemental figure 2a is likely meant to reference both S.Fig 2a and 2b.
- Line 118 - Supplemental figure 2b is likely intended to reference S.Fig 2c
- Line 124 - Typos hyphen after "on- and off turning"
- Line 154 - "a diversity of activities" should be "diverse activities"
- Line 154 - they use function and functionality in the same sentence.

- Line 159 – comma after “Western blot”
- Line 176 – Start sentence with “Therefore..”

Reviewer #2 (Remarks to the Author):

Impact/appropriateness

This paper describes a novel biochemical toolkit for modulating the activity of dCas9 via small molecules that alternatively stabilize or destabilize dCas9. The authors accomplish this by generating fusion proteins incorporating two systems: the Auxin-Inducible Degron (AID) and the ecDHFR destabilizing domain (ecDDD). Useful features of this control mechanism include rapid and reversible switching between stabilization states, and the ability to tune stability in an analog fashion by adjusting small molecule dosages. This concise report describes reagents that will certainly be of use to those seeking to construct gene expression functions using dCas9.

Recommendations

This manuscript could be improved by making a few changes.

1. General notes

Line 84: please suggest a plausible source of auxin in the culture medium. Is it not more likely that the IAA17dCas9 interacts modestly with α TIR1 even in the absence of auxin?

Line 148: please speculate as to why other nucleases were not well-regulated by auxin/AID

Lines 161-164: Similarly, can you provide more insight into why the orthogonal dCas9 proteins did not function well with the ecDDD degron tag?

Lines 169-171: Can you please speculate as to how the difference in function of the degron system between CHO and 293FT cells could be useful or detrimental?

The conclusion is quite brief. It would help the reader to better appreciate the utility of this work if the authors were to speculate as to some of the observations made in this study (as per previous comments), to compare the performance of this system against other reported systems (e.g., estradiol-regulated Cas9 developed by Savage et al., doi:10.1038/nbt.3528), and to explain how such a system could be used in a bit more specific detail (e.g., some examples).

2. Figures

Bar graphs in figures could be cleaned up. For example, consider removing background lines and boxes around panels. Bar graphs use different bar colors but there is no legend.

All figures from which a conclusion of significant difference is claimed should include statistical analysis (and indications of significance levels, e.g., p-values)

Moving the cartoon of the ecDHFR system to the main text figures (instead of the supplement) would be useful.

Figure 1b- For all Western blots, it would be helpful to indicate what antibody or epitope is being labeled in each panel (e.g., in the figure captions). Without this information, some features are difficult to interpret. In this figure, for example, what is the band around 135 kDa that is present without Cas9 but which is upregulated upon expression of dCas9? Is this explainable?

Figure 1c- the use of GFP is not clear unless the supplementary information is read; please add a

description in the figure caption about how GFP is expressed and how this control is to be interpreted.

Figure 3c- Please clarify in the caption that this experiment involves an effector-less AID-dCas9 expressing cell line (or add another row to the table indicating that this component is present in all cases). Although the use of a dual luciferase assay is mentioned, please clarify whether the data presented have been normalized to the firefly luciferase (the label, RLU, does not make this clear).

General comments re blots: It would be best to include (in the supplement) full Western blots for all images shown in excerpted form. It would be useful to show the entire high-exposure blot depicted as a blow-out in Supplementary Figure 3a.

3. Miscellaneous typos:

Line 48: need comma after "circuits"

Line 78: "HEK392FT cells" should be HEK293FT cells

Line 107: change Figure 2b to Figure 1b

Supplementary Figure 2- part C is not labeled (i.e., there is no "C" on the figure)

Reviewers' comments:

Reviewer #1 (Remarks to the Author):

The authors created a new Cas9 system capable of switching when induced using degron-tags. Using dCas9 and VP64 the CRISPR/Cas9 system can activate transcription of a gene of interest. This ability was coupled to degradation by auxin and the ecDHFR destabilizing domain. This new tool was able to reduce the activation by gRNA and dCas9-VP64 on genes upon addition of auxin. The ecDDD system was not compatible with dCas9, so the authors attached the destabilizing domain to MCP with limited success. Overall, auxin was used to rapidly modify levels of dCas9 in HEK293 and CHO cell lines. The manuscript provides details the use of protein tags to regulate the stability of dCas9 Effectors. In summary, this work is significant because it details an approach to improve the control of gene expression by dCas9, a widely-used technology. The use of multiple gene targets, Cas9 variants, and two simultaneous degrons represents a substantial body of work to support the manuscript. However, the impact of the work is greatly diminished by the lack of clarity in the manuscript and figures. The specific issues detailed below limit the reviewer's enthusiasm for the manuscript and may lessen most reader's appreciation for the work overall.

We like to thank this reviewer for their overall very positive and encouraging assessment of our study. We are sorry that some of her/his enthusiasm has been lost by the presentation of the work, but we believe that we have greatly improved this aspect in our revised manuscript.

Major comments for the authors

- On lines 84 and 85, the authors suggest that the small reduction in protein stability observed with the co-transfection of TIR may in the absence of auxin may be due to a "low level of auxin in the culture media". Since auxin is a plant hormone, this probably isn't the case.

We believe that a tiny amount of auxin is introduced via the fetal bovine serum (FBS) used in the culture medium; FBS is derived from cows which of course eat a lot of plant material. However, we agree that this is speculation and there could simply be a low level of interaction between the IAA17 tag and osTIR1 even in the absence of auxin. We have amended the text to include this possibility.

- On line 118, the authors detail how the addition of auxin reduced the fluorescent reporter readout to below background level, as shown in supplemental figure 2B. This doesn't seem to correlate with supplemental figure 2B, and it doesn't make sense as to why the degradation of a transactivator would reduce the luciferase readout to below background, as background should be the system without the transactivator.

We simply meant to state here that we could not detect fluorescent signal above detectable level, i.e. 'background level' was intended to indicate the endogenous (auto-) fluorescence levels of the cells. We have changed to text to 'below detectable level' to avoid confusion on this point.

- The authors need to comment on why there are such pronounced differences in the performance of the degron-tagged Cas9 orthologs in different cell lines (HEK293 and CHO). Since they are

discussing the use of this system to control gene programs, this may be something worth discussing, since other cell lines could benefit from the technology presented in this manuscript.

We are unclear why we observe differences between the HEK293 and CHO cells for some of the degron-tagged factors. This relates mainly to the stability of the AID-Cpf1 factors, and we think this could be due to differences in the expression levels of some auxiliary factors (e.g SCF complex subunits) in combination with reduced accessibility of the IAA17 tag at the N-termini of the Cpf1 proteins. We have added this point of discussion into the text.

- In Figure 1b- A western blot for dCas9 is shown, but no load control is shown. The authors should repeat the blot with a load control (b-actin), similar to figure 3B to make the claim that Cas9 co-transfection with tir1 effect degradation (lines 85-86).

This has been done, see new Figure 1b.

- Figure 2a- The load control for lane 6 (160 μ M) indicates a poorly loaded well and isn't appropriate for publication. The concentration is not absolutely necessary to make the authors claim, so either remove the lane or repeat.

This also has been repeated and is shown in new Figure 2a.

- Lines 130-138 need to be re-written for clarity because the results for figures 2d and 2e appear to be discussed twice.

These are two different experiments. In the first (figure 2d) protein levels following addition of auxin are sampled, while in the second (figure 2e) the reappearance of dCas9 after removal of auxin is measured.

- GFP load control needs to be better described. Is the control expressed from the same plasmid or co-transfected?

This is now better described in the figure legend. In short, a ubiquitous GFP expression construct was co-transfected and used as transfection and loading control.

- The figure data would be more clear if the cell types are clearly labeled and transient data is separated or at least distinguished from the stable cell lines.

We hope this is now clearer in the revised manuscript.

- Clarify when sgRNA mixes are used.

This has been done, e.g. lines 125-127 of the revised ms: 'targeting a number of endogenous genes via transfection of mixes of expression plasmids for 3 or 4 sgRNAs to sites within the same promoter region'.

- Add guide RNA description to the methods. Were these chosen at random, predicted by software, or candidates from a larger group?

A description has been added in the methods section (lines 271-275)

- Figure 2b and 2c are described as fast or rapid with no time in the presented data. The reference is possibly meant to describe slope or this reference is confused with 2d and 2e. The authors need to clarify the referencing to figures.

This has been amended in the text and figure legend.

- Figure 2d shows and change in protein without a load control. A load control is required to show changes in protein content.

The Western blot has been repeated with a loading control, as shown in new Figure 2.

- In the methods, there is no discussion of how the gRNAs are designed or of the mix gRNA experiments.

This has been added.

- Perform 2-sample t-test on fluorescence and qRTPCR data to test for significance.

Done and included in the figures.

Minor comments for the authors

- This manuscript reads as if it was a first draft, many revisions will likely be needed before the next submission.

We appreciate the time taken by the reviewer to provide such detailed feedback which has been very useful in our revisions and we agree that their comments have helped to improve our manuscript.

- Minimize subject-verb inversion for clearer writing.

Done

- Keep acronyms and naming consistent between figures, legends, and main text. One example is in supplementary figure 4c, pRL-TK and p8*BSL1-minP-GL4 are used. However, these acronyms can't be found anywhere else in the article and are never explained.

These construct names are now explained in the figure legend.

- Proofread for comma placement and check for comma splicing.

Done

- Break up large noun clusters - i.e. line 50-51: ...user-controlled temporal and dosage fine-tuning...

Done, see revised text from line 53 - 58.

- Discuss the CRISPR SAM system (Lines 176 through 181) in the introduction.

Done, see line 40-42 in the new text.

- It is unclear what you mean by "transactivation domain-less AID-dCas9". If it is just AID-dCas9 without a transactivation domain just call it AID-dCas9.

That is correct. We wrote it as such to make it clear that in this experiment we used AID-dCas9 without a transactivation domain, but we agree this unnecessarily doubles the information, hence we have changed the text to AID-dCas9 as suggested (see line 210)

- Make sure figures are provided in the same order that they are provided in the text. In this manuscript figures 1 a-c are discussed, followed by figure 2, then figures 1 d-f. Either split figure 1 into two separate figures or change the order of the writing so that figure 1 is completed before moving on to figure 2.

This should now be the case.

- It's unclear why GFP is included in figure 1c.

A ubiquitous GFP expression construct was co-transfected in this experiment so it could be used as transfection and loading control.

- Place subfigures in figure 1f on the same axes.

The genes that were used to measure upregulation by our degron-tagged dCas9 activator (AID-dCas9-PR) and its abrogation by the presence of auxin were chosen to represent a wide range of inducibilities, hence they cannot be presented on the same axis. We have however changed the arrangement of the panels in figure 1 so that the subfigures are now on the same horizontal plane.

- Line 107 which figure are the authors referencing?

Thanks for the alert. This should be figure 1d (not 2b as mentioned in the text: now fixed).

- Line 119 incorrect statement.

This sentence has been rewritten.

- “Multi-directional” is not a clear term to use in the title of the paper.

We have changed this term in the title to ‘multidimensional’, which, we agree, better reflects the content of the manuscript.

- Misspelling of “HEK293” on line 78.

Thanks, corrected.

- ecDDD is first defined on line 65 as “ecDHFR destabilizing domain”, but then the term “ecDHFR DD” is used on line 156. It is recommended that the authors pick one term and stick with it.

Thanks, we have changed the text to be more consistent and use ecDDD as much as possible.

- On lines 171 and 172, supplemental figure 5 is referenced for the AID and ecDHFR degron tagged sets, but the AID set is in supplemental figure 3.

Thanks, corrected.

- For lines 185-187, the only dCas9 observed in the table in Fig 3c is the one that has a VP64, yet they refer to a “transactivation domain-less AID-dCas9”.

The cell line used for the transfections in this experiment is stably expressing AID-dCas9, i.e. AID-dCas9 is present in all samples. In one of the samples dCas9-VP64 (without degron tag) was additionally transfected to serve as a positive control. We agree however that this may not be obvious to the reader and hence we have added a row to the table to indicate the presence of AID-dCas9 in all samples.

- On lines 238 and 239, they switched up the letter of the figures that they are referencing.

Corrected.

- Figure 3A needs to be clearer, because it's not apparent that the MCP is associating with the MS2 domains, or that the IAA17 would interact with the auxin and osTIR1.

We have improved the schematic drawing to take these points into account and hope that the new diagram is clearer (which is now Figure 4a).

- In the first line of the table in Figure 3 C they didn't include the AID tag in the name.

See response to query 'For lines 185-187';

- In Supplemental Figure 4 C, columns 3 and 4 appear to be the same in regards to the table.

Thanks for pointing this out: these were essentially duplicate transfections. We have now removed one of these to avoid confusion.

- Improve the consistency of figure style e.g. some have titles on top, others inside the plot.

Thanks, we have done this as much as possible.

- Figures should be referenced as Fig.1 or figure1 in accordance with the journal style. For example, in line 107 the authors write (figure 2b) and in line 117 the authors write (fig. 1d).

Done.

- All figures with RT-qPCR data- Y-axis should be titled " $\Delta\Delta\text{ct}$ fold change" or similar

Done, we have opted for 'fold upregulation'.

- Figure 1a- Label on osTIR1 (the white text) is not easily visible

Agreed, we have now moved this label so that it now appears as black on the white background.

- Figure 1d re-title the figure (remove underscores in all titles)

Okay, done.

- Figure 2a - Auxin concentration is given the unit uM instead of μM

Thanks, well-spotted, we have corrected this.

- Figure 3a This figure would be more helpful to the reader if the schematic depicted only the protein interactions, but with greater detail.

We hope the revised schematic drawing (which is now Figure 4a) is clearer.

- Line 40-44 – one long sentence. Should maybe break it up somewhere.

Done. See line 43-47 in the new text.

- Line 48 – should have a comma after “circuits”

Well-spotted and now corrected.

- Lines 50-64 – these lines are really dense and have a lot of words that I needed to look up (i.e. degnon, ecDHFR, TIR1).

We have inserted a short explanation of the term ‘degnon’ with a reference to a review paper on this subject. The other terms are explained elsewhere in the introduction.

- Line 63 - co-expression of the proteins in a mammalian system may lead to drawbacks.

This can indeed be a problem, hence we put in the effort to make the ‘all-in-one’ construct with the IAA17 degnon plus osTIR1 gene in the same plasmid.

- Lines 65-72 – a better explanation paragraph than the previous one. Easier to follow

Thanks.

- Line 76 – FRT-site containing pcDNA5 (?)

We have changed this to: ‘we generated a construct incorporating an FRT-site followed by a promoter-less hygromycinR gene cassette (Figure 1a)’

- Line 78- Change HEK392 to HEK293

Done.

- Line 107- Figure 2b is discussed before figure 1d,e,f.

Thanks for the alert. This should have read Figure 1d instead of 2b. It has now been corrected.

- Line 110 - Supplemental figure 2a is likely meant to reference both S.Fig 2a and 2b.

Correct, and corrected in the text.

- Line 118 - Supplemental figure 2b is likely intended to reference S.Fig 2c

Correct and fixed.

- Line 124 - Typos hyphen after “on- and off turning”

Okay.

- Line 154 – “a diversity of activities” should be “diverse activities”

Changed to ‘diverse range of activities’.

- Line 154 – they use function and functionality in the same sentence.

Thanks, functionality changed to versatility.

- Line 159 – comma after “Western blot”

Added.

- Line 176 – Start sentence with “Therefore..”

Thanks for the suggestion. We have opted for ‘We therefore ...’.

Reviewer #2 (Remarks to the Author):

Impact/appropriateness

This paper describes a novel biochemical toolkit for modulating the activity of dCas9 via small molecules that alternatively stabilize or destabilize dCas9. The authors accomplish this by generating fusion proteins incorporating two systems: the Auxin-Inducible Degron (AID) and the ecDHFR destabilizing domain (ecDDD). Useful features of this control mechanism include rapid and reversible switching between stabilization states, and the ability to tune stability in an analog fashion by adjusting small molecule dosages. This concise report describes reagents that will certainly be of use to those seeking to construct gene expression functions using dCas9.

We would like to thank to this reviewer for their brief but appreciative assessment. We have taken their comments on board and hope to have improved our manuscript using the suggested changes. Below we address each of their points.

Recommendations

This manuscript could be improved by making a few changes.

1. General notes

Line 84: please suggest a plausible source of auxin in the culture medium. Is it not more likely that the IAA17dCas9 interacts modestly with oTIR1 even in the absence of auxin?

See reviewer 1 comment: Text has been amended to accommodate this alternative possibility.

Line 148: please speculate as to why other nucleases were not well-regulated by auxin/AID

We are currently unclear why there appear to be some difference in the observed responses to the AID between the dCas9 and Cpf1 variants. One potential explanation is that due to structural differences between the proteins the degron tag may be less easily accessible to its interacting factors (eg. TIR1). Also, we have also observed that expression of the AID-dCas9 and AID-Sa_dCas9 proteins from stably transformed cell lines significantly increases the performance of the AID system, and this is likely to apply to the other variants as well.

Lines 161-164: Similarly, can you provide more insight into why the orthogonal dCas9 proteins did not function well with the ecDDD degron tag?

Similarly, we are unclear about this, but in our experience the ecDDD seems to be a less powerful degradation tag, with its activity potentially also influenced by protein structure. While a short linker was included between the ecDDD and the rest of the protein we have not experimented with variations in the length or nature of this linker. Furthermore, we think it is likely that generation of stable cell lines for the ecDDD tagged dCas9/Cpf1 proteins would improve the system.

Lines 169-171: Can you please speculate as to how the difference in function of the degron system between CHO and 293FT cells could be useful or detrimental?

Differences between the cell lines are more pronounced for some of the dCas9/Cpf1 factors than others. This knowledge can be useful in the design of experiments that combine multiple orthogonal dCas9/Cpf1 factors so that they can be selected for optimal degradation or stabilisation potential.

The conclusion is quite brief. It would help the reader to better appreciate the utility of this work if the authors were to speculate as to some of the observations made in this study (as per previous comments), to compare the performance of this system against other reported systems (e.g., estradiol-regulated Cas9 developed by Savage et al., doi:10.1038/nbt.3528), and to explain how such a system could be used in a bit more specific detail (e.g., some examples).

We have extended the discussion to include the suggested topics as well as some further related issues.

2. Figures

Bar graphs in figures could be cleaned up. For example, consider removing background lines and boxes around panels. Bar graphs use different bar colors but there is no legend.

Thanks for this suggestion; we have done this and provided explanation of the bar colors in the legends.

All figures from which a conclusion of significant difference is claimed should include statistical analysis (and indications of significance levels, e.g., p-values)

Done, see revised figures.

Moving the cartoon of the ecDHFR system to the main text figures (instead of the supplement) would be useful.

This diagram does not relate to the main text figures, hence we prefer to leave it as part of supplementary figure 5, but the gist of it is also represented as part of the diagram of figure 4a (new figure numbers)

Figure 1b- For all Western blots, it would be helpful to indicate what antibody or epitope is being labeled in each panel (e.g., in the figure captions). Without this information, some features are difficult to interpret. In this figure, for example, what is the band around 135 kDa that is present without Cas9 but which is upregulated upon expression of dCas9? Is this explainable?

We believe this to be a non-specific band that is picked up by the Cas9 antibody. The level of this band is not influenced by dCas9 expression. We believe that the small degree of upregulation that seemed to be present in the original figure 1b was an artefact, and we have not seen any change in its level on other blots, including in the repeated blot that forms the new figure 1b.

Figure 1c- the use of GFP is not clear unless the supplementary information is read; please add a description in the figure caption about how GFP is expressed and how this control is to be interpreted.

GFP was co-transfected from a ubiquitous GFP expression construct as a transfection and loading control. We have now added this information in the figure legend.

Figure 3c- Please clarify in the caption that this experiment involves an effector-less AID-dCas9 expressing cell line (or add another row to the table indicating that this component is present in all cases). Although the use of a dual luciferase assay is mentioned, please clarify whether the data presented have been normalized to the firefly luciferase (the label, RLU, does not make this clear).

We have added a row in the table to indicate that an effector-less AID-dCas9 is present in all samples, expressed from a stably integrated construct. We have also added an explanation about the dual luciferase assay in which firefly luciferase reporter activity is normalised to the internal

control of a ubiquitously expressed renilla luciferase.

General comments re blots: It would be best to include (in the supplement) full Western blots for all images shown in excerpted form. It would be useful to show the entire high-exposure blot depicted as a blow-out in Supplementary Figure 3a.

We will place images of the full Western blots in the online data repository that will become associated with the paper.

3. Miscellaneous typos:

Line 48: need comma after "circuits"

Inserted.

Line 78: "HEK392FT cells" should be HEK293FT cells

Thanks, repaired.

Line 107: change Figure 2b to Figure 1b

Done.

Supplementary Figure 2- part C is not labeled (i.e., there is no "C" on the figure)

This has now been added.